# Thymoquinone Inhibits Growth of Acute Myeloid Leukemia Cells through Reversal *SHP-1* and *SOCS-3* Hypermethylation: In Vitro and In Silico Evaluation

**DOI:** 10.3390/ph14121287

**Published:** 2021-12-09

**Authors:** Futoon Abedrabbu Al-Rawashde, Muhammad Farid Johan, Wan Rohani Wan Taib, Imilia Ismail, Syed Ahmad Tajudin Tuan Johari, Belal Almajali, Abdullah Saleh Al-wajeeh, Mansoureh Nazari Vishkaei, Hamid Ali Nagi Al-Jamal

**Affiliations:** 1School of Biomedicine, Faculty of Health Sciences, Universiti Sultan Zainal Abidin (UniSZA), Kuala Nerus 21300, Terengganu, Malaysia; futoonrawashdeh1001@gmail.com (F.A.A.-R.); wanrohani@unisza.edu.my (W.R.W.T.); imilia@unisza.edu.my (I.I.); bel_basss@yahoo.com (B.A.); 2Department of Haematology, School of Medical Sciences, Universiti Sains Malaysia, Kubang Kerian 16150, Kelantan, Malaysia; faridjohan@usm.edu.my; 3Centre for Research in Infectious Diseases and Biotechnology (CeRIDB), Faculty of Medicine, Universiti Sultan Zainal Abidin, Kuala Terengganu 20400, Terengganu, Malaysia; syedtajudin@unisza.edu.my; 4Anti-Doping Lab Qatar, Doha P.O. Box 27775, Qatar; a_alwajeeh@yahoo.com; 5School of Pharmacy, University of 17 August 1945, Jakarta 14350, Indonesia; nazarimansoure@gmail.com

**Keywords:** thymoquinone, hypomethylation, *SHP-1*, *SOCS-3*, *FLT3*-ITD, AML

## Abstract

Epigenetic silencing of tumor suppressor genes (TSGs) plays an essential role in cancer pathogenesis, including acute myeloid leukemia (AML). All of *SHP-1*, *SOCS-1*, and *SOCS-3* are TSGs that negatively regulate JAK/STAT signaling. Enhanced re-expression of TSGs through de-methylation represents a therapeutic target in several cancers. Thymoquinone (TQ) is a major component of *Nigella sativa* seeds with anticancer effects against several cancers. However, the effects of TQ on DNA methylation are not entirely understood. This study aimed to evaluate the ability of TQ to re-express *SHP-1*, *SOCS-1*, and *SOCS-3* in MV4-11 AML cells through de-methylation. Cytotoxicity, apoptosis, and cell cycle assays were performed using WSTs-8 kit, Annexin V-FITC/PI apoptosis detection kit, and fluorometric-red cell cycle assay kit, respectively. The methylation of *SHP-1*, *SOCS-1*, and *SOCS-3* was evaluated by pyrosequencing analysis. The expression of *SHP-1*, *SOCS-1*, *SOCS-3*, *JAK2*, *STAT3*, *STAT5A*, *STAT5B*, *FLT3*-ITD, *DNMT1*, *DNMT3A*, *DNMT3B*, *TET2*, and *WT1* was assessed by RT-qPCR. The molecular docking of TQ to *JAK2*, *STAT3*, and *STAT5* was evaluated. The results revealed that TQ significantly inhibited the growth of MV4-11 cells and induced apoptosis in a dose- and time-dependent manner. Interestingly, the results showed that TQ binds the active pocket of *JAK2*, *STAT3*, and *STAT5* to inhibit their enzymatic activity and significantly enhances the re-expression of *SHP-1* and *SOCS-3* through de-methylation. In conclusion, TQ curbs MV4-11 cells by inhibiting the enzymatic activity of JAK/STAT signaling through hypomethylation and re-expression of JAK/STAT negative regulators and could be a promising therapeutic candidate for AML patients.

## 1. Introduction

Acute myeloid leukemia (AML) is a hematopoietic stem cell malignancy characterized by distinct genetic and epigenetic abnormalities. FMS-like tyrosine kinase 3 (*FLT3*) with internal tandem duplication (*FLT3*-ITD) and Janus Kinase 2 (*JAK2*) mutations are frequent genetic events in AML [1]. Aberrant DNA methylation is one of the most typical causes of AML initiation and progression [1,2]. AML is associated with a high frequency of mutations in genes involved in DNA methylation, including DNA methyltransferases (DNMTs); *DNMT1*, *DNMT3A*, and *DNMT3B* [3], and DNA demethylases; ten-eleven translocation 2 (*TET2*) [4] and Wilms Tumor 1 (*WT1*) [5,6].

Epigenetic silencing of tumor suppressor genes (TSGs) by DNA hypermethylation has a critical role in the development of leukaemia, including AML [7,8]. Unlike genetic dysregulation, the epigenetic alterations are reversible, making them attractive targets for anticancer therapy.

Hyperactivation of the Janus tyrosine kinase-signal transducer and activator of transcription (JAK/STAT) signaling plays an essential role in the pathogenesis of AML [9]. Src homology 1 domain-containing protein tyrosine phosphatase (*SHP-1*), suppressor of cytokine signaling-1 (*SOCS-1*), and suppressor of cytokine signaling-3 (*SOCS-3*) are TSGs that negatively regulate the growth-promoting signaling, such as JAK/STAT signaling [9,10,11]. Epigenetic silencing of *SHP-1* and *SOCS-3* by promotor hypermethylation has been identified in hematological malignancies, including AML [10,12,13]. Epigenetic silencing of *SOCS-1* has also been reported in myelodysplastic syndrome [14] and multiple myeloma [15]. However, the different methylation status of CpG islands of *SOCS-1* has been identified in AML [16,17,18]. The restoration of *SHP-1*, *SOCS-1*, and *SOCS-3* expression by epigenetic modulating agents such as 5-azacytidine (5-Aza) was associated with significant inhibition of *JAK2*/*STAT3* and *STAT5* signaling in AML leukemia cells [8,12]. However, the response rate of the conventional DNA hypomethylating agents is low [19,20]. Therefore, new treatments that efficiently modify aberrant epigenetic mechanisms are vitally needed.

Overexpression of *FLT3* due to ITD mutation plays an essential role in the survival and proliferation of AML cells [21]. Targeting *FLT3*-ITD represents an attractive therapeutic target for AML patients with this mutation [22]. *FLT3* tyrosine kinase inhibitors (TKIs) have been used in clinical trials. However, the response rates to these TKIs were temporary [23,24]. Natural phytochemical compounds have induced anti-cancer effects in *FLT3*-ITD overexpressing cells [25], suggesting that phytochemical compounds can inhibit the expression of the ITD-mutated *FLT3* gene.

Natural phytochemical compounds are excellent and safe alternatives for cancer therapy. These compounds may target and modulate the genetic expression by interacting with the genetic and epigenetic mechanisms [26]. Thymoquinone (TQ) (2-methyl-5-isopropyl-1,4-benzoquinone) is a phytochemical compound extracted from *Nigella sativa* seeds that can modulate cancer cells epigenetic mechanisms, including histone acetylation or deacetylation and DNA methylation or de-methylation [27,28]. However, the anti-leukemic activities of TQ and its effect on DNA methylation are still not thoroughly investigated.

The current study hypothesized that *SHP-1*, *SOCS-1*, and *SOCS-3* lose their tumor suppression function in AML due to epigenetic silencing, and TQ could re-express these TSGs through de-methylation. For this purpose, the effect of TQ on the methylation status and the expression of *SHP-1*, *SOCS-1* and, *SOCS-3* were studied in *FLT3*-ITD positive MV4-11 AML cells. The effect of TQ on the expression of *FLT3*-ITD, *JAK2*, *STAT3*, *STAT5A*, *STAT5B*, *DNMT1*, *DNMT3A*, *DNMT3B*, *TET2*, and *WT1* genes, cell proliferation, apoptosis, and cell cycle progression as well as the molecular docking of TQ to *JAK2*, STAT 3 and *STAT5* was also evaluated.

## 2. Results

### 2.1. Thymoquinone Inhibits Cell Proliferation in MV4-11 Cells

The results showed that cell viability significantly decreased in a dose- and time-dependent phenomenon. At 24 h, 48 h, and 72 h, the IC_50_s were calculated from the dose-cell viability percentage curve; 7.8 ± 1.6 μM, 5.5 ± 1.3 μM, and 3.8 ± 0.96 μM, respectively (Figure 1A). The viability of MV4-11 cells after 24 h, 48 h, and 72 h of exposure to TQ were evaluated using the trypan blue exclusion method to confirm the growth inhibition effects at TQ IC_50_ values. The trypan blue exclusion assay results also showed that TQ inhibited the proliferation of MV4-11 cells in a dose- and time-dependent manner (Figure 1B).

### 2.2. Thymoquinone Induces Dose and Time-Dependent Apoptosis in MV4-11 Cells

The MV4-11 cells treated with IC_50_ values of TQ revealed that TQ induced apoptosis in a dose- and time-dependent manner (Figure 2A,B). Total apoptosis was observed in 49.51% of MV4-11 cells after treatment with 7.8 μM TQ for 24 h, 51.21% of MV4-11 cells demonstrated total apoptosis after treatment with 5.5 μM for 48 h, and 44.26% of MV4-11 cells showed total apoptosis after treatment with 3.8 μM for 72 h. Additionally, 72 h of treatment with 7.8 μM TQ induced the most significant rise in the percentage of cells with total apoptosis (91.13%).

### 2.3. Thymoquinone Induces Cell Cycle Arrest at G0/G1 Phase in MV4-11 Cells

Treatment with TQ significantly increased the G0/G1 population, whereas the G2/M population was decreased in a dose- and time-dependent manner. The G0/G1 population showed the most pronounced increase after 72 h of 7.8 μM TQ treatment (80.57%) compared to untreated cells (59.23%). Meanwhile, both S and G0 populations were not affected (Figure 3A,B).

### 2.4. Thymoquinone Creates a Balance in the Expression of the Regulators of DNA Methylation Genes in MV4-11 Cells

The expression of the regulators of DNA methylation was investigated in MV4-11 cells before and after treatment with TQ. The results of RT-qPCR analysis showed that the expression of *DNMT1*, *DNMT3A*, and *DNMT3B* was significantly decreased by 1.8-fold (*p* < 0.001), 2.9-fold (*p* < 0.001), and 1.3-fold (*p* = 0.0003), respectively, compared to untreated cells. The expression of *TET2* and *WT1* was significantly increased by 1.9-fold (*p* < 0.001) and 2.8-fold (*p* < 0.001), respectively, compared to untreated cells (Figure 4A).

### 2.5. Thymoquinone Induces SHP-1 and SOCS-3 Promoter Hypomethylation in MV4-11 Cells

Quantitative pyrosequencing was performed to investigate DNA methylation levels in untreated MV4-11 cells, TQ-treated MV4-11 cells, 5-Aza-treated MV4-11 cells, methylated bisulfite-converted DNA control, unmethylated bisulfite-converted DNA control, and unmethylated bisulfite-unconverted DNA control. The results of DNA methylation levels of four CpG sites in the *SHP-1* promoter 2 region, seven CpG sites in the *SOCS-1* promoter region, and three CpG sites in the *SOCS-3* promoter region were compared in MV4-11 cells before and after treatment with TQ. Pyrosequencing results showed a decrease in the methylation level of CpG islands in the promoter 2 region of *SHP-1* gene in TQ-treated MV4-11 cells (7.5%) compared to (9.3%) untreated MV4-11 cells (*p* = 0.037) (Table 1, Figure 5). The results also showed a decrease in the methylation level of CpG islands in the promoter region of *SOCS-3* gene in TQ-treated MV4-11 cells (2.0%) compared to (4.7%) in untreated MV4-11 cells (*p* = 0.041) (Table 2, Figure 6). On the other hand, the promoter region of *SOCS-1* in both treated and untreated MV4-11 cells was not methylated (Table 3, Figure 7).

### 2.6. Thymoquinone Enhances Re-Expression of SHP-1, SOCS-1, and SOCS-3 in MV4-11 Cells

The expression of JAK/STAT-negative regulator genes in MV4-11cells was analyzed by RT-qPCR. After 48 h of 5.5 μM of TQ treatment, the mRNA levels of *SHP-1*, *SOCS-1*, and *SOCS-3* were significantly increased (*p* < 0.001). The expression of *SHP-1* increased 5-fold after TQ treatment relative to untreated cells. The expression of *SOCS-1* and *SOCS-3* also increased by 2.34 and 2.12-fold, respectively (Figure 4B).

### 2.7. Thymoquinone Downregulates FLT3-ITD and JAK/STAT Signaling

MV4-11 cells were analyzed for different mRNA expression patterns of *FLT3*-ITD, *JAK2*, *STAT3*, *STAT5A*, and *STAT5B* before and after treatment with TQ using RT-qPCR. The expression of *FLT3*-ITD and *JAK2* was significantly decreased by 8-fold (*p* < 0.001) and 5.3-fold (*p* < 0.001), respectively, compared to untreated cells (Figure 4C). The expression of *STAT3*, *STAT5A*, and *STAT5B* was also significantly decreased after TQ treatment by 4.2-fold, 3.2-fold, and 5.6-fold, respectively, compared to that in untreated cells (*p* < 0.001) (Figure 4C).

### 2.8. Thymoquinone Binds the Active Pocket of JAK2, STAT3, and STAT5 

The actual docked conformation of *JAK2*, *STAT3*, and *STAT5* with the active conformation of each ligand, TQ and 5-Aza, clearly showed that numerous potential interactions were present (Figure 8 and Figure 9). The comparison of free binding energy and Ki of both ligands after interaction with studied proteins is presented in Table 4.

The interactions between TQ and *JAK2* showed hydrogen bond with TYR913, pi lone pair with LEU905, pi donor hydrogen bond with GLN906, and two alkyl bonds with LYS912 and ILE910 with a free binding energy of −5.99 Kcal/mol (Figure 8C). The interactions of the positive control, 5-Aza, with *JAK2* showed six hydrogen bonds with LYS903, GLN906, TYR913, HIS907, and ILE910 and alkyl bond with LYS912 with a free binding energy of −6.37 Kcal/mol (Figure 9B). The interaction between TQ and *JAK2* showed higher free binding energy and less affinity than 5-Aza.

The interaction between TQ and *STAT3* demonstrated two hydrogen bonds with LEU508 and TRP1479, pi sulphur bond with MET1482, pi sigma bond with VAL1507, and four alkyl bonds with LEU1478, VAL1504, PHE1549, and TRP1562 with a free binding energy of −6.68 Kcal/mol (Figure 8D). The interactions of the positive control, 5-Aza, with *STAT3* showed four hydrogen bonds with *STAT3* in CYS1259, ARG1262, VAL1136, and GLN1141, pi cation bond with ARG1246, and alkyl bond with LEU1263 (Figure 9C). TQ showed less free binding energy with *STAT3* than 5-Aza (−6.52 Kcal/mol) and more affinity toward *STAT3*; this can be due to more stability that happened in the active pocket of the protein after interaction with TQ.

The interaction of TQ with *STAT5* demonstrated four hydrogen bonds with LEU1259, GLN1258, GLN1139, and ARG1242, two alkyl bonds with CYS1255 and ALA1254, and pi pi T shaped with TRP1262 with a free binding energy of −6.04 Kcal/mol (Figure 8E). The positive control, 5-Aza, demonstrated three hydrogen bonds with TYR1356, GLU164 and ASN1391 of *STAT5*, and an alkyl bond with LYS279 of *STAT5* with a free binding energy of −6 Kcal/mol (Figure 9D). The free binding energy of TQ after interaction with *STAT5* was slightly lower than 5-Aza, so it showed a slightly better affinity towards *STAT5*. This can be due to two hydrogen bonds interacting with ketonic oxygen at the *para* position of the cyclohexane in the chemical structure of TQ; this gives more stability in the active pocket of *STAT5* compared with 5-Aza.

## 3. Discussion

Aberrant methylation of TSGs is involved in the pathogenesis of several cancers [7]. The expression of TSGs, such as the JAK/STAT-negative regulator genes, is silenced due to hypermethylation of the CpG islands in promoter regions, leading to the malignant transformation of normal hematopoietic cells and the development of leukemia [7,8]. Different levels of DNA methylation have been implicated in the development and prognosis of AML [8]. Re-expression of silenced TSGs by inhibiting DNA methylation is an important therapeutic target for AML treatment. However, the therapeutic efficiency of the conventional DNA hypomethylating agents is limited [19,20]. Therefore, novel agents that efficiently target aberrant epigenetic mechanisms are highly needed.

Phytochemical compounds, such as curcumin, were found to inhibit the DNA methylation process [29]. TQ is a phytochemical compound, and few researchers have investigated its potential as a modulator of the DNA methylation process [27]. To investigate TQ as a therapeutic agent for AML treatment, it is essential to identify genes and signal transduction pathways involved in TQ-induced AML cells’ inhibition. While previous studies have revealed various mechanisms behind the anti-leukemia effects of TQ [30,31], in this study, we identify hypomethylation of JAK/STAT-negative regulator genes as previously unstudied epigenetic alterations that mediate TQ’s anti-leukemic activities in AML.

The present study showed that TQ exhibited an antiproliferative effect in MV4-11 cells in a time- and dose-dependent manner. The increment of TQ concentration after 24 h of treatment showed significant inhibition of MV411 cells proliferation up to 71% (Figure 1A). Thymoquinone also showed a time-dependent growth inhibitory effect in MV4-11 cells. This was demonstrated by the significant increase of inhibitory impact by 3 μM from 18% at 24 h to 64% at 72 h (Figure 1A–C). The trypan blue exclusion assay revealed that TQ at 7.8 μM decreased cell proliferation to 50% after 24 h of treatment. The cell proliferation was significantly reduced to 21% and 13% after 48 and 72 h of incubation with TQ, respectively (Figure 1B). These findings suggested that TQ inhibited MV4-11 cell proliferation in a dose- and time-related manner. These results are consistent with the previous reports, in which TQ showed a dose-proportional inhibitory effect in MV4-11, Kasumi 1 AML cells [22], and HL60 AML cells [32].

DNMTs regulate the DNA methylation process and have a critical role in normal hematopoiesis [3]. Dysregulation and mutations of *DNMT1*, *DNMT3A*, *DNMT3B*, *TET2* and *WT1* genes are associated with the development of AML [3,4,6]. A proper balance in the expressions of DNMTs, *TET2*, and *WT1* is essential to regulate the DNA methylation process [33]. In this study, we have evaluated the effect of TQ on the expression of *DNMT1*, *DNMT3A*, *DNMT3B*, *TET2*, and *WT1* in MV4-11 AML cells. Our findings showed that TQ significantly decreased *DNMT1*, *DNMT3A*, and *DNMT3B* mRNA levels in MV4-11 cells (Figure 4A). On the other hand, the results revealed that TQ enhanced the expression of both *WT1* and *TET2* in MV4-11 cells (Figure 4A). The results of the present study indicate that the downregulation of *DNMT1*, *DNMT3A*, and *DNMT3B* was associated with the upregulation of *WT1* and *TET2*, which potentially reduce DNA methylation in MV4-11 cells. In agreement with our results, TQ significantly decreased the expression of *DNMT1* and *DNMT3A* in the primary blast cells from AML patients, ML-1, Kasumi-1, and MV4-11 AML cells [22]. TQ also has decreased the expression of *DNMT1*, *DNMT3A*, and *DNMT3B* in Jurkat T-cell acute lymphoblastic leukemia cells [34]. TQ also has shown upregulation of *TET2* in HECV human vascular endothelial cells [35]. In addition, curcumin treatment downregulated *DNMT1* expression in MV4-11 AML cells [36].

The pathogenesis of AML involves a constitutive activation of JAK/STAT signaling [10,12,13]. JAK/STAT signaling is inhibited by SOCS and SHP protein families [37]. Both *SHP-1* and *SOCS-3* have shown epigenetic silencing due to promoter hypermethylation in haematological malignancies, including AML [10,12,13]. However, the methylation status of the promoter region of *SOCS-1* has not been well studied in AML. In our previous study, TQ increased the expression of *SHP-1*, *SOCS-1*, and *SOCS-3* genes in HL60 AML cells [32], supporting the idea that TQ could target the epigenetic silenced JAK/STAT-negative regulator genes in AML cells. To better understand TQ’s DNA hypomethylating activity, we examined its effect on the methylation level of CpG islands in the promoter regions of *SHP-1*, *SOCS-1*, and *SOCS-3* in MV4-11 cells.

Pyrosequencing analysis showed that TQ induced hypomethylation of *SHP-1* promotor 2 region from 9.3% in untreated MV4-11 cells to 7.5% in TQ-treated cells (*p =* 0.037) (Table 1, Figure 5). Additionally, the results indicated that TQ induced hypomethylation of the promoter region of *SOCS-3* from 4.7% in untreated MV4-11 cells to 2% in TQ-treated cells (*p =* 0.041) (Table 2, Figure 6). These results indicate that TQ-induced hypomethylation of *SHP-1* and *SOCS-3* is attributed to its effect in creating appropriate balance in the expression of the regulators of DNA methylation. TQ also induced hypomethylation of *SHP-1* promotor 2 region and *SOCS-3* promoter region in K562 leukemia cells (Data not shown). These findings are consistent with what has been reported in which TQ decreased DNA methylation in MV4-11 cells and the primary blast cells from AML patients, resulting in inhibition of cell proliferation [22]. Additionally, TQ has been reported to induce hypermethylation of the *TWIST1* promoter region, which inhibits the growth of breast cancer cells and cervical cancer cells [28,38]. Furthermore, TQ has been found to alter the epigenetic status of Jurkart cells by downregulating many key epigenetic players and upregulating TSGs previously identified as epigenetically silenced in a variety of cancers, including leukaemia [34].

Previous studies have identified different methylation statuses of *SOCS-1* in AML [16,17,18]. This study showed that the CpG sites in the *SOCS-1* promoter are unmethylated in MV4-11 cells (Table 3, Figure 7), indicating that they are not involved in the epigenetic regulation of *SOCS-1* transcription, thus is not suitable for methylation studies. Consistent with our findings, *SOCS-1* promoter region was unmethylated in AML patients [16,17]. On the other hand, *SOCS1* promoter was methylated in the relapsed group of AML patients [39], and only 25% of the M5 subtype of AML patients showed a methylation status of *SOCS1* promoter region with a significant variation in *SOCS-1* methylation among various cytogenetic subgroups [18].

In this study, we further evaluated the effect of TQ-induced hypomethylation on the expression of *SHP-1* and *SOCS-3* in MV4-11 cells. The results showed that TQ significantly increased *SHP-1* and *SOCS-3* mRNA levels in MV4-11 cells (Figure 4B). The effect of TQ on the expression of *SOCS-1* was also investigated, and the results showed that TQ significantly increased *SOCS-1* mRNA level. The unmethylated status of *SOCS-1* leads us to hypothesize that the mechanism underlying TQ-induced re-expression of *SOCS-1* is different from that behind the re-expression of *SHP-1* and *SOCS-3*. These findings are consistent with previous studies in which *SHP-1* and *SOCS-3* were re-expressed in leukemia cells after 5-Azacytidine treatment [8] and support our previous findings in which TQ increased the expression of *SHP-1*, *SOCS-1*, and *SOCS-3* in HL60 AML cells [32].

Constitutive activation of *STAT3*, *STAT5A*, and *STAT5B* is confirmed variably in AML cells [40,41,42]. Therefore, genes involved in JAK/*STAT3* and *STAT5* signaling are considered as an important molecular target for AML treatment. In this study, we found that TQ reduced the mRNA levels of *JAK2*, *STAT3*, *STAT5A*, and *STAT5B* in MV4-11 cells (Figure 4C). These results are consistent with previous findings in which scutellarin treatment has decreased the mRNA level of *JAK2* in hepatocellular carcinoma cells [43], TQ treatment has downregulated *STAT3* in gastric cancer cells [44], and dasatinib treatment has downregulated *STAT5A* and *STAT5B* in K562 CML cells [45]. JAK/STAT signaling pathway was also inhibited by curcumin treatment through downregulating JAK/STAT downstream target genes in retinoblastoma cells [46].

The findings of this study also demonstrated that TQ binds the active pocket of *JAK2*, *STAT3*, and *STAT5* proteins with a negative free binding energy and proper affinity compared with the positive control (5-Aza) selected in this study (Figure 8 and Figure 9, and Table 4), which led to the inhibition of *JAK2*, *STAT3* and *STAT5* enzymatic activity in MV4-11 cells. In agreement with our results, TQ was found to bind the catalytic site of *DNMT1* and inhibit its enzymatic activity in MV4-11 cells [22].

Hyperactivation of JAK/STAT signaling is associated with increased expression and altered signaling through growth factor receptors in AML stem cells, including *FLT3* [21,47]. As a result of the *FLT3*-ITD mutation, MV4-11 AML cells have constitutive activation of *FLT3* and its downstream targets, including *STAT3* and *STAT5*, leading to uninhibited cell proliferation and impairment of differentiation and apoptosis [48]. Inhibition of *FLT3* expression significantly inhibits JAK/STAT signaling in AML [49]. In this study, we examined the effect of TQ on the expression of *FLT3*-ITD in MV4-11 cells. We found that TQ treatment significantly impaired the expression of *FLT3*-ITD in MV4-11 cells (Figure 4C). Our results confirm previous findings indicating that the inhibition of *FLT3* signaling represents one of several molecular mechanisms underlying TQ-induced cancer cell growth inhibition [22].

Transcriptional targets of the JAK-activated STAT family include genes involved in regulating cell survival, proliferation, and apoptosis [50,51]. In the present study, we evaluated the effect of TQ on the apoptosis of MV4-11 cells. The analysis revealed that TQ caused a concentration-dependent rise of early and late apoptosis in MV4-11 cells, with 91.13% total apoptosis after 72 h of incubation with 7.8 µM TQ, compared to 72.04% and 49.76% total apoptosis at 5.5 μM and 3.8 µM TQ, respectively. Moreover, TQ also induced a significant time-dependent effect in apoptosis, as illustrated in Figure 2A,B. These results agreed with other studies that showed that TQ enhanced apoptosis in C91, HuT-102, CEM, and Jurkat human T-cell leukaemia cells [30], and MV4-11 AML cells [22].

Thymoquinone has been shown to have anticancer properties against several types of leukaemia by inducing DNA damage and G0/G1 cell cycle arrest [30,52]. In line with the literature, this study found that TQ considerably blocked cell cycle transition in MV4-11 cells at the G0/G1 phase in a time- and dose-dependent manner (Figure 2A,B). At a concentration of 7.8 µM, TQ increased the percentage of arrested MV4-11 cells in the G0/G1 phase to 80.57%.

In conclusion, TQ mediated anti-leukemia effects on MV4-11 AML cells by creating a balance in the expression of the epigenetic regulator genes through downregulating *DNMT1*, *DNMT3A*, and *DNMT3B* and upregulating *TET2* and *WT1*, resulting in a reduction in DNA methylation levels of *SHP-1* and *SOCS-3* and restoration of their expression. In addition, TQ inhibited the enzymatic activity of *JAK2*, *STAT3*, and *STAT5* by binding to their active pockets with a negative free binding energy. Consequently, TQ induced proliferation inhibition, apoptosis induction, and cell cycle arrest. 

## 4. Materials and Methods

### 4.1. Cell Culture 

*FLT3*-ITD positive MV4-11 AML cells were obtained from Elabscience Biotechnology Co., Ltd., (Wuhan, China). The cells were maintained in Iscove’s Modified Dulbecco Medium (IMDM) supplemented with 10% (*v*/*v*) fetal bovine serum (FBS) and 1% (*v*/*v*) penicillin-streptomycin. The cell culture was maintained in a humidified incubator supplied with 5% carbon dioxide (CO_2_) at 37 °C and incubated until 70% confluency. IMDM, FBS, and penicillin-streptomycin solution were purchased from Elabscience Biotechnology Co., Ltd., (Wuhan, China).

### 4.2. TQ and 5-Aza Treatment 

TQ (>98% pure) was obtained from Sigma Aldrich (Sigma Aldrich Corp., Louis, MO, USA). TQ was prepared with 2% dimethyl sulfoxide (DMSO) at a stock concentration of 30 mM and stored at −80 °C, then 5.5 µM working concentration was prepared with IMDM medium at the onset of treatment. 5-Aza was purchased from Solarbio life sciences (Beijing, China). The 5-Aza was prepared at a stock solution of 10 mM with 2% DMSO and stored at −80 °C, and the working concentration of 2.3 µM was prepared with IMDM medium.

### 4.3. Cytotoxicity Assay 

Cell counting water-soluble tetrazolium salt-8 (WST-8) kit (Nacalai Tesque Inc., Kyoto, Japan) was used to determine the cytotoxicity of TQ at various concentrations for 24 h, 48 h, and 72 h. The MV4-11 cells were seeded in 96-well culture plates at a density of 5 × 10^3^ viable cells/100 μL/well and pre-incubated at 37 °C in a 5% CO_2_ incubator for 24 h. Subsequently, 10 μL of varied TQ working concentrations (3 μM, 6 μM, 9 μM, and 12 μM) were added on the designated wells, while the blank (culture media only) and control (untreated cells) wells received 10 μL of culture media. The plates were then incubated for 24 h, 48 h, and 72 h at 37 °C in a 5% CO_2_ incubator. Next, 10 μL of WST-8 solution was added to each well of the plate and placed in a similarly conditioned incubator for four hours. After that, the absorbance was measured at 450 nm (calibration wavelength: 600 nm) by a microplate reader (Infinite M200, Tecan, Männedorf, Switzerland). Four independent experiments were performed. The potency of TQ was expressed by half-maximal inhibitory concentration (IC_50_) values, which were calculated from the dose-cell viability percentage curve with blank absorbance value subtracted to remove background absorbance.

### 4.4. Cell Viability Assay 

The viability of MV4-11 cells was measured using the trypan blue exclusion assay. The MV4-11 cells were seeded in triplicate in a 96-well plate (1 × 10^4^ cells/mL) and treated with 7.8 μM, 5.5 μM, and 3.8 μM of TQ for 24 h, 48 h, and 72 h, respectively. Then, 10 µL of cell suspension from each incubation was mixed with 10 µL of trypan blue solution (0.2% in PBS), and the viable (unstained) and dead (blue) cells were counted using a hemocytometer (Reichert, Buffalo, NY, USA). Four independent experiments were performed. The viability of TQ-treated cells was expressed as a percentage of viable cells relative to untreated ones (100%).

### 4.5. Apoptosis Assay

The cell apoptosis rate was measured using the annexin V-FITC/propidium iodide (PI) apoptosis detection kit (Nacalai Tesque Inc., Kyoto, Japan). The TQ-treated MV4-11 cells (7.8 μM, 5.5 μM, and 3.8 μM of TQ for 24 h, 48 h, and 72 h, respectively) were centrifuged, and the pellet was washed twice with PBS. The cells were then resuspended in an annexin V binding buffer with a final cell concentration of 1 × 10^6^ cells/mL. For flow cytometric preparation, 100 μL of the cell suspensions were incubated with 5 μL of annexin V-FITC conjugate and 5 μL of PI solution for 15 min at room temperature (RT) in the dark. Then, 400 μL of annexin V binding buffer was added to the solution and applied to the CytoFLEX flow cytometer (Beckman Coulter, Brea, CA, USA). For each sample, 10,000 events were acquired, and positive FITC and/or PI cells were quantified and analyzed by CytExpert for CytoFLEX Acquisition and Analysis Software (Beckman Coulter, USA). Three independent experiments were performed and the data were presented as histograms.

### 4.6. Cell Cycle Analysis Using Flow Cytometry

Cell cycle analysis was performed using the fluorometric–red cell cycle assay kit (Elabscience Biotechnology Co., Ltd., Wuhan, China). The MV4-11 cells treated with IC_50_ concentrations of TQ (7.8 μM, 5.5 μM, and 3.8 μM for 24 h, 48 h, and 72 h, respectively) were centrifuged and resuspended gently in PBS before counting. Next, cells of 5 × 10^5^ cells/tube concentration were prepared, centrifuged, and resuspended in 0.3 mL PBS. The cell suspension was then fixed with 1.2 mL iced absolute ethanol, oscillated followed by incubation at −20 °C overnight. Thereafter, the fixed cells were centrifuged, and the pellet was resuspended in 1 mL PBS and incubated at RT for 15 min. The solution was then centrifuged again, and the pellet was resuspended in 100 μL RNase A reagent and continued to be incubated for 30 min in a 37 °C water bath. Subsequently, 400 μL of PI staining solution was added, thoroughly mixed, and incubated for 30 min at 4 °C. Following that, a flow cytometer (Beckman Coulter, USA) was used to analyse the cells. Three independent experiments were performed. The percentage of cell cycle phase distribution was calculated by ModFit LT 4.1 software (Beckman Coulter, USA).

### 4.7. RNA Extraction

Total cellular RNA was extracted from both TQ-treated (5.5 µM for 48 h) and untreated MV4-11 cells using ReliaPrep™ RNA Cell Miniprep System RNA extraction kit (Promega, Madison, WI, USA) following the manufacturer’s protocol. The purity and concentration of the extracted RNA were measured by Nanodrop-Photometer (Implen, Weslake Village, CA, USA).

### 4.8. Quantitative Reverse Transcriptase PCR (RT-qPCR) 

GoTaq^®^ 2-Step RT-qPCR System kit (Promega, USA) was used to reverse transcribe 100 ng of RNA into cDNA. The expression of *SHP-1*, *SOCS-1*, *SOCS-3*, *FLT3*-ITD, *JAK2*, *STAT3*, *STAT5A*, *STAT5B*, *DNMT1*, *DNMT3A*, *DNMT3B*, *TET2*, and *WT1* was analyzed using SYBR Green-based GoTaq 2-Step RT–qPCR System kit (Promega, USA). Each PCR reaction was performed in triplicates whereby each reaction contained 10 μL of GoTaq^®^ qPCR Master Mix (2×), 0.2 μL of CXR reference dye, 1 μL of forward (20×) and reverse primers (20×), 6.8 μL nuclease-free water, and 2 μL of cDNA template [2 μL of nuclease-free water were added to the no template control (NTC)]. The cycling conditions for the RT-qPCR reactions were as follows: activation of GoTaq^®^ DNA Polymerase, 2 min at 95 °C; denaturation step, 40 cycles at 95 °C for 15 s; Annealing and extension, 40 cycles at 60 °C for 1 min. RT-qPCR was analyzed using Applied Biosystem StepOnePlus™ thermocycler (Applied Biosystems, Foster City, CA, USA). The primers sequences used for PCR amplification are listed in Table 5. Data were analyzed by StepOne Software v2.3 (ABI step one plus, New York, NY, USA). The relative expression levels of the genes were determined using RT-qPCR and the 2^−∆∆Cq^ method [53] after normalization to the endogenous reference β-actin, and the results were presented as fold changes.

### 4.9. DNA Extraction and Bisulfite Treatment

MV4-11 cells were treated with 5.5 µM TQ for 48 h. The demethylating agent, 5-Aza was used as positive control, and the cells were treated with 2.3 µM of 5-Aza for 48 h. Bisulfite-converted completely methylated, or unmethylated human genomic DNAs and bisulfite-unconverted completely unmethylated human genomic DNA were used as controls (EpiTect PCR Control DNA Set, Qiagen, Hilden, Germany). Total cellular DNA was extracted from both treated and untreated MV4-11 cells using Wizard^®^ Genomic DNA Purification Kit (Promega, USA) according to the manufacturer’s protocol. The concentration and purity of the extracted DNA were assessed by Nanodrop-Photometer (Implen, Weslake Village, CA, USA). Tow micrograms of DNA were treated with bisulfite using the EpiTect Bisulfite Kit (Qiagen, Hilden, Germany) according to the manufacturer’s instructions.

### 4.10. Primer Design for Pyrosequencing Assay

Quantitative DNA methylation analysis of bisulfite-treated DNA was performed by pyrosequencing of the promotor 2 region of *SHP-1* and the promotor regions of *SOCS-1* and *SOCS-3* genes. For each gene, we selected the CpG islands in the promoter area, flanking the transcription start site at 5′UTR. Three to seven CpG sites were studied for each particular CpG island. Primers for PCR amplification and pyrosequencing were designed using PyroMark Assay Design Software version 2.0 (Qiagen, Hilden, Germany) and synthesized by PyroMark Custom Assay (Qiagen, Hilden, Germany). The reverse primers were biotin labelled, allowing for the production of biotinylated PCR product necessary for the pyrosequencing reactions. The analyzed promotor regions and the corresponding primers sequences used for pyrosequencing analysis for each gene are presented in Table 6.

### 4.11. Docking of Thymoquinone 

Python language was downloaded from www.python.com (accessed on 1 July 2021), Molecular graphics laboratory (MGL) tools was downloaded from http://mgltools.scripps.edu (accessed on 1 July 2021) and AutoDock4.2 was downloaded from http://autodock.scripps.edu (accessed on 1 July 2021), Bio Via draw was downloaded from http://accelrys.com (accessed on 1 July 2021), Discovery studio visualizer 2017 was downloaded from http://accelrys.com (accessed on 1 July 2021), and Chem3D was downloaded from https://acms.ucsd.edu (accessed on 1 July 2021) [63].

The three-dimensional crystal structure of anticancer targets *JAK2* with PDB ID: 3KCK, *STAT3* with PDB ID: 3CWG, and *STAT5* with PDB ID: 1Y1U were selected and downloaded from Protein Data Bank (www.rvcsb.org/pdb ((accessed on 1 July 2021) (Figure 9A). The complexes bound to the receptor molecule, all the non-essential water molecules and heteroatoms were deleted and ultimately, hydrogen atoms were added to the target receptor molecule using Argus Lab [64].

The structures of TQ and 5-aza were available with the identified structure of crystallography. Pubchem was used to make sdf format and converted to PDB format using Pymol and further used for docking studies. The starting structures of the proteins were prepared using AutoDock tools [65]. Water molecules were deleted, and polar hydrogen and Kollman charges were added to the protein starting structure. Grid box was set with the size of 126 × 126 × 126Å with the grid spacing of 0.375 Å at the binding site. The starting structure of TQ was constructed using BioVia draw, and 5-Aza was selected as positive control. The structures of TQ and 5-Aza were provided from the Pubchem website, Gasteiger charges were assigned into optimized ligands using Autodock Tools. One hundred docking runs were conducted with a mutation rate of 0.02 and a crossover rate of 0.8 [62]. The population size was set to use 250 randomly placed individuals. Lamarckian Genetic algorithm was used as the searching algorithm with a translational step of 0.2 Å, a quaternion step of 5 Å, and a torsion step of 5 Å. The most populated and lowest free binding energy was selected as the final result [66].

### 4.12. Statistical Analysis

Data analyses were conducted using the Statistical Package for Social Sciences (SPSS) version 25 (SPSS Inc., Chicago, IL, USA). The data between TQ-treated and untreated MV4-11 cells were analyzed using Paired samples *t*-test and Wilcoxon signed-rank test. The data between groups were analyzed using repeated measures ANOVA with Bonferroni correction with the significant level set at *p* < 0.05, *p* < 0.01, and *p* < 0.001.

## 5. Conclusions

The anti-leukemic effect of TQ was investigated on MV4-11 AML cells. TQ potentially inhibited the cell proliferation, enhanced apoptosis, and arrested cell cycle of MV4-11 AML cells at G0/G1 phase. These effects could be attributed to TQ mediated hypomethylation of *SHP-1* and *SOCS-3* through downregulating *DNMT1*, *DNMT3A*, and *DNMT3B* genes resulting in re-expression of *SHP-1* and *SOCS-3* genes. Moreover, TQ inhibited the enzymatic activity of *JAK2*, *STAT3*, and *STAT5* by binding to their active pockets. TQ also downregulated the expression of *FLT3*-ITD, *JAK2*, *STAT3*, and *STAT5*. These findings suggest that TQ could be a future therapeutic candidate for the treatment of AML patients through DNA hypomethylation of JAK/STAT negative regulators.

## Figures and Tables

**Figure 1 pharmaceuticals-14-01287-f001:**
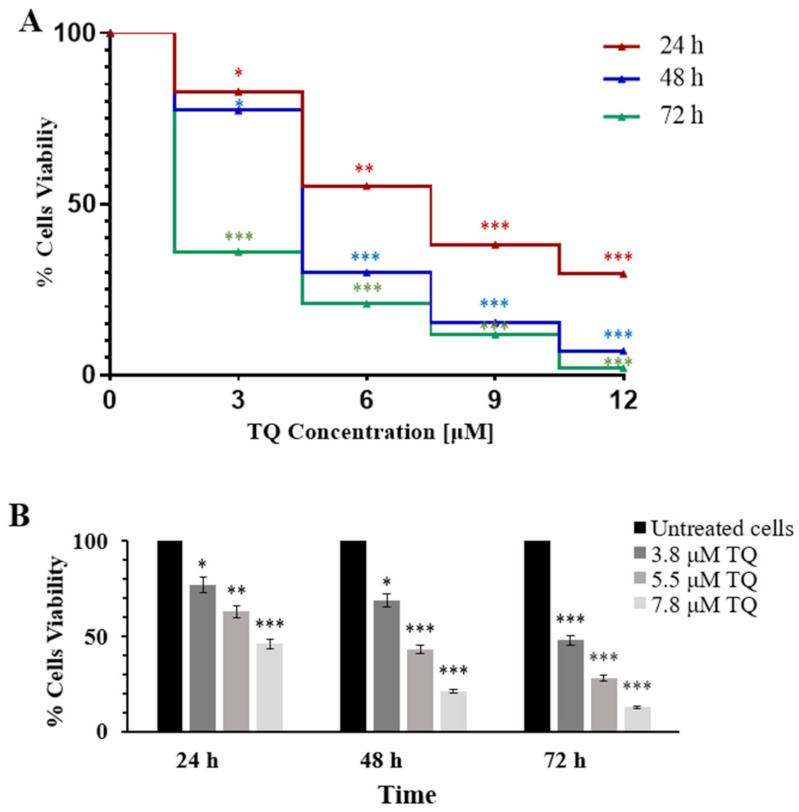
Dose- and time-dependent growth inhibitory effect of TQ in MV4-11 cells. (**A**) Cytotoxicity of TQ in MV4-11cells was determined by cell counting WST-8 assay after 24 h, 48 h, and 72 h. The IC_50_ values were 7.8 μM, 5.5 μM, and 3.8 μM, respectively. (**B**) The viability of MV4-11 cells was confirmed by trypan blue exclusion assay after exposure to IC_50_ concentrations of TQ for 24 h, 48 h, and 72 h. Data were presented as a percentage relative to the untreated cells. The Paired samples *t*-test was applied, and the results are stated as mean ± SD (n = 4), where * *p* < 0.05, ** *p* < 0.01, *** *p* < 0.001 are significant versus untreated control cells.

**Figure 2 pharmaceuticals-14-01287-f002:**
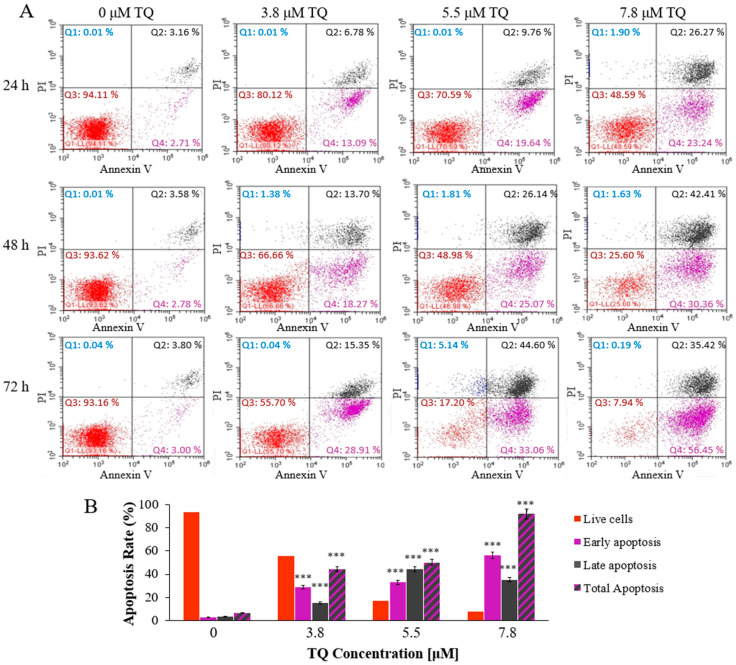
Dose- and time-course effects of TQ on apoptosis in MV4-11 cells. (**A**) Cells were treated with 7.8 μM, 5.5 μM, and 3.8 μM of TQ at 24 h, 48 h, and 72 h. The annexin V-FITC and PI-stained MV4-11 cells were assessed for apoptosis by flow cytometry. (**B**) The graph indicates the apoptosis rate in MV4-11 cells after 72 h of exposure to IC_50_ values. The number of total apoptotic cells is presented as a percentage relative to the total cell numbers. The mean apoptosis before and after treatment and between the four groups showed significant difference when tested with repeated measures ANOVA. Data are representative of three independent experiments and expressed as means ± SD. Where *** *p* < 0.001 is significant versus untreated control cells.

**Figure 3 pharmaceuticals-14-01287-f003:**
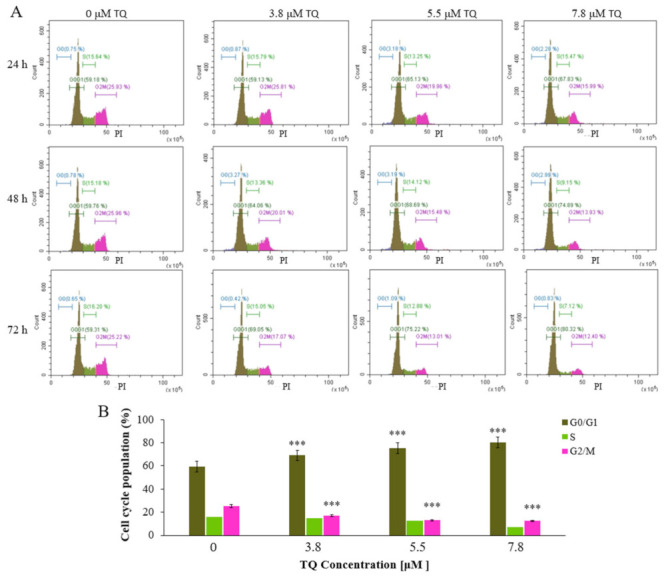
Flow cytometry histograms for cell cycle analysis of MV4-11cells. (**A**) Cells were treated with TQ (7.8 μM, 5.5 μM, and 3.8 μM for 24 h, 48 h, and 72 h). Thymoquinone significantly increased the G0/G1 population in MV4-11 cells in a time- and dose-dependent manner. (**B**) The graph indicates the cell cycle distribution in MV4-11 cells after 72 h of exposure to IC_50_ values. Data are representative of three independent experiments and expressed as mean ± SD. Where *** *p* < 0.001 is significant versus untreated control cells. The mean of the G0/G1 population at all treatment groups showed significant differences when tested with repeated measures ANOVA (*p* < 0.001).

**Figure 4 pharmaceuticals-14-01287-f004:**
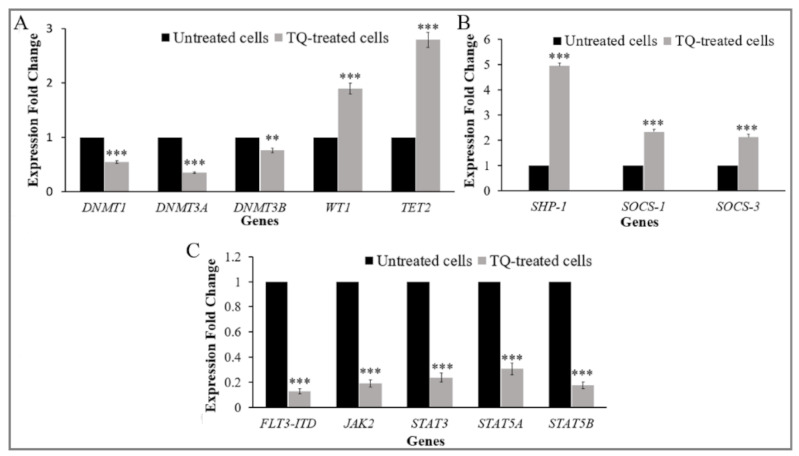
Effects of TQ on gene expression in MV4-11cells. Cells were treated with 5.5 μM of TQ for 48 h. (**A**) The graph illustrates the downregulation of *DNMT1*, *DNMT3A*, and *DNMT3B* and the up-regulation of *TET2* and *WT1*. (**B**) The graph shows the up-regulation of *SHP-1*, *SOCS-1*, and *SOCS-3*. (**C**) The graph shows the downregulation of *FLT3*-ITD, *JAK2*, *STAT3*, *STAT5A*, and *STAT5B*. Values are stated as median (interquartile range) (n = 3); where ** *p* < 0.01, *** *p* < 0.001 are significant versus untreated control cells.

**Figure 5 pharmaceuticals-14-01287-f005:**
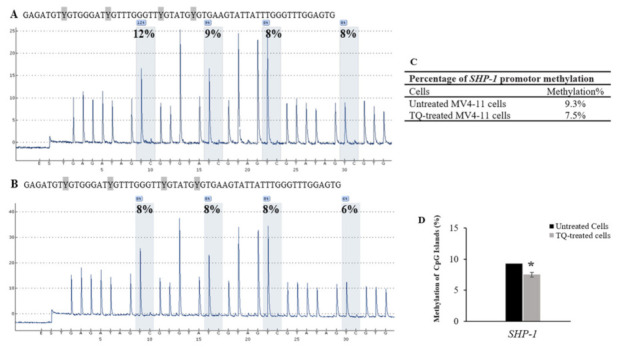
Effect of TQ on the methylation of *SHP-1* promoter 2 region in MV4-11 cells. (**A**) Pyrosequencing pyrograms for the *SHP-1* promoter 2 region methylation status in untreated MV4-11 cells and (**B**) TQ-treated MV4-11 cells. The percentages shown above each “C” base are the methylation level of the CpG site. (**C**) Percentages of *SHP-1* methylation in MV4-11 cells. (**D**) The bar graph shows hypomethylation of *SHP-1* in MV4-11 cells after TQ treatment. Repeated measures ANOVA was applied and the values are stated as mean ± SD; where * *p* < 0.05 is significant versus untreated control cells.

**Figure 6 pharmaceuticals-14-01287-f006:**
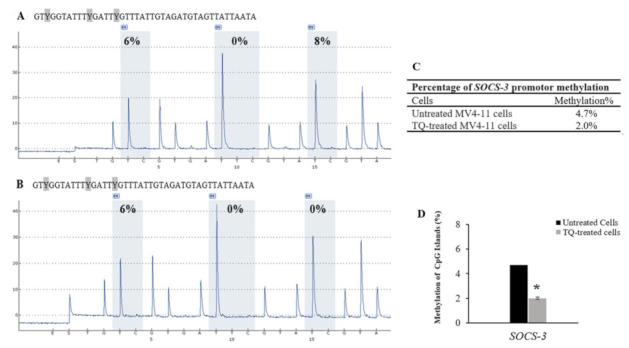
Effect of TQ on the methylation of *SOCS-3* promoter region in MV4-11 cells. (**A**) Pyrosequencing pyrograms for the methylation status of *SOCS-3* promoter region in untreated MV4-11 cells and (**B**) TQ-treated MV4-11 cells. The percentages shown above each “C” base are the methylation level of the CpG site. (**C**) Percentages of *SHP-1* methylation in MV4-11 cells. (**D**) The bar graph shows hypomethylation of *SOCS-3* in MV4-11 cells after TQ treatment. Repeated measure ANOVA was applied, and the values are stated as mean ± SD; where * *p* < 0.05 is significant versus untreated control cells.

**Figure 7 pharmaceuticals-14-01287-f007:**
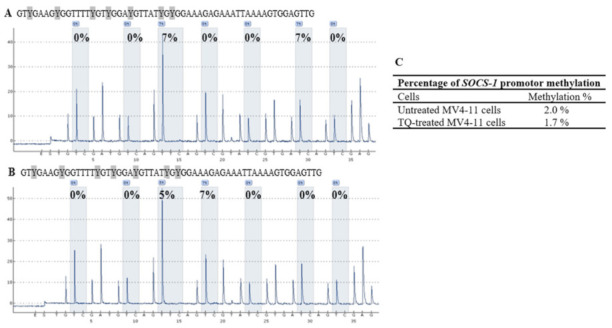
Pyrosequencing pyrograms for the methylation status of *SOCS-1* promoter in (**A**) untreated MV4-11 cells; (**B**) TQ-treated MV4-11 cells. The percentages shown above each “C” base are the methylation level of the CpG site. The percentages shown above each “C” base are the methylation level of the CpG site. (**C**) Percentages of *SOCS-1* methylation in MV4-11 cells.

**Figure 8 pharmaceuticals-14-01287-f008:**
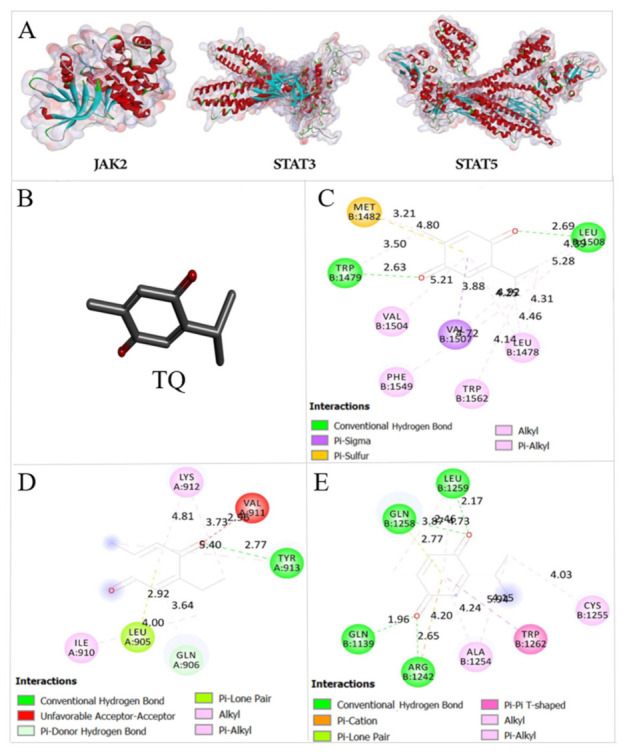
(**A**) *JAK2* with PDB ID: 3KCK, *STAT3* with PDB ID: 3CWG, and STAT5 with PDB ID: 1Y1U from the protein data bank. (**B**) Crystallography structure of TQ. The interactions between TQ and the studied proteins (**C**) *JAK2*, (**D**) *STAT3*, and (**E**) STAT5.

**Figure 9 pharmaceuticals-14-01287-f009:**
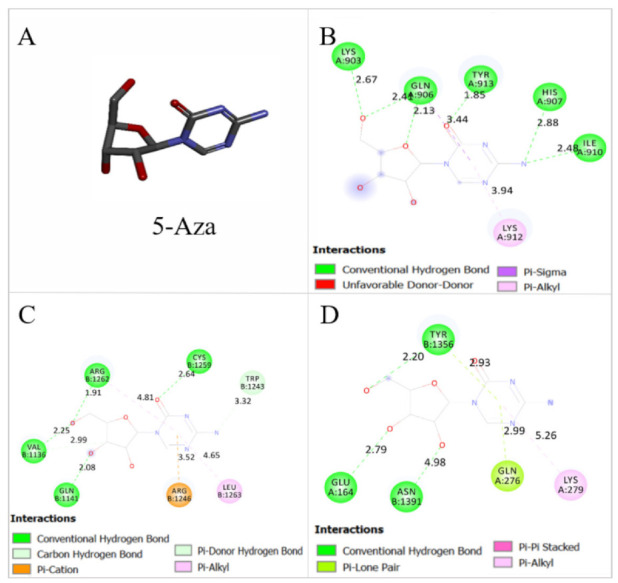
(**A**) Crystallography structure of the positive control, 5-Aza. The interactions between the 5-Aza and the studied proteins (**B**) *JAK2*, (**C**) *STAT3*, and (**D**) STAT5.

**Table 1 pharmaceuticals-14-01287-t001:** Percentage of methylation of CpG sites in the promoter region of *SHP-1* gene.

Cells	CpG-1	CpG-2	CpG-3	CpG-4	Mean ± SD
Untreated MV4-11 cells	12	9	8	8	9.3 ± 1.8
TQ-treated MV4-11 cells	8	8	8	6	7.5 ± 1.0
5-Aza-treated MV4-11 cells	13	8	0	0	5.2 ± 2.1
Unmethylated DNA control	6	0	0	0	1.5 ± 0.4
Methylated DNA control	95	88	100	84	91.8 ± 15.7
Unmethylated Bisulfite-unconverted DNA control	0	0	0	0	0

**Table 2 pharmaceuticals-14-01287-t002:** Percentage of methylation of CpG islands in the promoter region of *SOCS-3* gene.

Cells	CpG-1	CpG-2	CpG-3	Mean ± SD
Untreated MV4-11 cells	6	0	8	4.7 ± 1.1
TQ-treated MV4-11 cells	6	0	0	2.0 ± 0.7
5-Aza-treated MV4-11 cells	0	0	0	0
Unmethylated DNA control	0	0	0	0
Methylated DNA control	89	96	100	95.0 ± 15.5
Unmethylated Bisulfite- unconverted DNA control	0	0	0	0

**Table 3 pharmaceuticals-14-01287-t003:** Percentages of methylation of CpG islands in the promoter region of *SOCS-1* gene.

*SOCS-1*	CpG-1	CpG-2	CpG-3	CpG-4	CpG-5	CpG-6	CpG-7	Mean ± SD
Untreated MV4-11 cells	0	0	7	0	0	7	0	2.0 ± 0.4
TQ-treated MV4-11 cells	0	0	5	7	0	0	0	1.7 ± 0.9
5-Aza-treated MV4-11 cells	0	0	0	0	0	0	0	0
Unmethylated DNA control	5	0	8	6	0	6	0	3.6 ± 1.3
Methylated DNA control	82	86	100	89	93	100	79	89.9 ± 22.3
Unmethylated Bisulfite-unconverted DNA control	0	0	0	0	0	0	0	0

**Table 4 pharmaceuticals-14-01287-t004:** Free binding energy (FBE) of TQ and 5-Aza after interaction with studied proteins.

	FBE (Kcal/mol)		Ki	
	TQ	5-Aza	TQ	5-Aza
** *JAK2* **	−5.99	−6.37	40.79 μM	21.51 μM
** *STAT3* **	−6.68	−6.52	12.76 μM	16.68 μM
**STAT5**	−6.04	−6.0	37.64 μM	40.21 μM

**Table 5 pharmaceuticals-14-01287-t005:** Primer sequences for RT-qPCR assay.

Genes	Accession Numbers	Primer Sequence (5′-3′)	Reference
*SHP-1*	NC_000012.12	Forward: GCCTGGACTGTGACATTGAC Reverse: ATGTTCCCGTACTCCGACTC	[43]
*SOCS-1*	NC_000016.10	Forward: GACGCCTGCGGATTCTAC Reverse: AGCGGCCGGCCTGAAAG	[54]
*SOCS-3*	NC_000017.11	Forward: GACCAGCGCCACTTCTTCAC Reverse: CTGGATGCGCAGGTTCTTG	[55]
*FLT3*-ITD	NC_000013.11	Forward: ACGCTTGGAAGCAGGAGAT Reverse: CACAAGGCTGCCCTCTAGTT	[56]
*JAK2*	NC_000009.12	Forward: TGTCTTACCTCTTTGCTCAGTGGCG Reverse: CAATGACATTTTCTCGCTCGACAGC	[57]
*STAT3*	NC_000017.11	Forward: GATTGACCAGCAGTATAGCCGCTTC Reverse: CTGCAGTCTGTAGAAGGCGTG	[58]
*STAT5A*	NC_000017.11	Forward: GAAGCTGAACGTGCACATGAATC Reverse: GTAGGGACAGAGTCTTCACCTGG	[45]
*STAT5B*	NC_000017.13	Forward: AGTTTGATTCTCAGGAAAGAATGT Reverse: TCCATCAACAGCTTTAGCAGT	[45]
*DNMT1*	NC_000019.10	Forward: TAT CCG AGG AGG GCT ACC Reverse: TAA GCA TGA GCA CCG TTCT	[59]
*DNMT3A*	NC_000002.12	Forward: GGA GGA CCGAAA GGA CGG A Reverse: CCC CATTGG GTA ATA GCTCTG AG	[59]
*DNMT3B*	NC_000020.11	Forward: GAG ATC AGA GGC CGA AGA T Reverse: CTG TCA AGT CCT GTG TGTAG	[59]
*TET2*	NC_000004.12	Forward: ACGCTTGGAAGCAGGAGAT Reverse: CACAAGGCTGCCCTCTAGTT	[60]
*WT1*	NC_000011.10	Forward: CAGGCTGCAAAAGAGATATTTTAAGCT Reverse: GAAGTCACACTGGTATGGTTTCTCA	[61]
*β-actin*	NC_000071.7	Forward: CTGGCACCCAGGACAATG Reverse: GCCGATCCACACGGAGTA	[62]

**Table 6 pharmaceuticals-14-01287-t006:** Bisulfite-converted sequences, PCR and pyrosequencing primer sequences for pyrosequencing assays. CpG sites are highlighted in grey.

Assay Name	Analyzed Bisufite Sequence	Type of Primer	Primer Sequence (5′-3′)	Amplicon Length
** *SHP-1* **	GAGATGTYGTGGGATYGTTTGGGTTYGTATGYGTGAAGTATTATTTGGGTTTGGAGTG	PCR-Forward	GGAGGGTTGAGTAAAAGTAGTTGG	94 bp
		PCR-Reverse	Biotin_ACACTCCAAACCCAAATAATACTT	
		Sequencing	TTGGTGGAGGAGGGA	
** *SOCS-1* **	GTYGAAGYGGTTTTYGTYGGAYGTTATYGYGGAAAGAGAAATTAAAAGTG GAGTTG	PCR-Forward	AGGGTTTAGAAGAGAGGGAAATA	85 bp
		PCR-Reverse	Biotin_CCCCCAACTCCACTTTTAATT	
		Sequencing	GAAGAGAGGGAAATAGG	
** *SOCS-3* **	GTYGGTATTTYGATTYGTTATTGTAGATGTAGTTATTAA TA	PCR-Forward	GGAGGTTTTGGGTTTGGTATTTAGTAA	207 bp
		PCR-Reverse	Biotin_AAACCCTTCCCCAAATCTCATAAAT	
		Sequencing	ATTTAGTAAAGTTGTGGTTTGAG	

## Data Availability

Data sharing contain in this article.

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
