# Peer review of "Thymoquinone Inhibits Growth of Acute Myeloid Leukemia Cells through Reversal *SHP-1* and *SOCS-3* Hypermethylation: In Vitro and In Silico Evaluation"

_pharmaceuticals, 2021, doi:10.3390/ph14121287_

Round 1
Reviewer 1 Report
In the article "Thymoquinone Inhibits Growth of Acute Myeloid Leukemia 2
Cells Through Reversal SHP-1 and SOCS-3 Hypermethylation: 3
In Vitro and Silico Evaluation," the authors have studied the effect of Thymoquinone on inhibiting the enzymatic activity of JAK/STAT signaling through hypomethylation and re-expression of JAK/STAT negative regulator.
In this study, the authors have used only one FLT3-ITD cell line, and the authors need to use one more cell line with the FLT3 mutation to check the effect on the methylation status. Also, have the authors tested the effect of TQ on any non-tumor cell line?
The authors need to check that figure 4, 4a, and 4c is the same graph, and the authors should change figure 4c according to the figure description.
In figure 4a, the descriptive results of RT-qPCR analysis of DNMT1, DNMT3A, and DNMT3B have shown 1.8, 2.9-and 1.3-fold change, but the bar graph does not show the same.
Author Response
Date:04 December 2021
Dear Editor
Thank you for giving us the opportunity to submit the revised manuscript entitled “Thymoquinone Inhibits Growth of Acute Myeloid Leukemia Cells Through Reversal SHP-1 and SOCS-3 Hypermethylation: In Vitro and In Silico Evaluation” for publication in Pharmaceuticals Journal.
We appreciate your and reviewers valuable comments on the manuscript. Please be informed that we have intensively revised the manuscript and highlighted the changes in the manuscript track changes.
Reviewer 1
In the article "Thymoquinone Inhibits Growth of Acute Myeloid Leukemia Cells Through Reversal SHP-1 and SOCS-3 Hypermethylation: In Vitro and Silico Evaluation," the authors have studied the effect of Thymoquinone on inhibiting the enzymatic activity of JAK/STAT signaling through hypomethylation and re-expression of JAK/STAT negative regulator.
Comment 1. In this study, the authors have used only one FLT3-ITD cell line, and the authors need to use one more cell line with the FLT3 mutation to check the effect on the methylation status. Also, have the authors tested the effect of TQ on any non-tumor cell line?
Authors Response:
Dear reviewer, your suggestion is highly appreciated and please be informed that our team has already investigated the effect of TQ on the methylation status of the same genes in another leukemia cell line and the results were similar in both cell lines. Therefore, the comment has considered and the statement ”TQ also induced hypomethylation of SHP-1 promotor 2 region and SOCS-3 promoter region in K562 leukemia cells (Data not shown)” has added.
In this study we have focused on the antileukemia effect of thymoquinone on acute myeloid leukemia MV4-11 cells. However, the effect of thymoquinone on activated peripheral blood mononuclear cells (PBMC) non-tumor cells had been examined and reported previously, in which the cells were more resistant to thymoquinone treatment compared to cancer cells suggesting that thymoquinone does not effect on non-malignant cells; Dergarabetian, E. M., Ghattass, K. I., El-Sitt, S. B., Al-Mismar, R. M., El-Baba, C. O., Itani, W. S., ... & Gali-Muhtasib, H. U. (2013). Thymoquinone induces apoptosis in malignant T-cells via generation of ROS. Frontiers in bioscience (Elite edition), 5, 706-719.
Comment 2. The authors need to check that figure 4, 4a, and 4c is the same graph, and the authors should change figure 4c according to the figure description.
Authors Response:
Thank you for your comment and please be informed that the mistake in Figure 4 has corrected.
Comment 3. In figure 4a, the descriptive results of RT-qPCR analysis of DNMT1, DNMT3A, and DNMT3B have shown 1.8, 2.9-and 1.3-fold change, but the bar graph does not show the same.
Authors Response:
Dear reviewer, your suggestion is highly appreciated. The results of RT-qPCR analysis showed a significant DOWN-REGULATION of DNMT1, DNMT3A, and DNMT3B (expression fold change= 0.55, 0.35, and 0.76, respectively), which means that mRNA levels were DECREASED by 1.8, 2.9 and 1.3 times compared to the control cells (expression fold change =1).

Reviewer 2 Report
Reviewer 1
Pharmaceuticals-1471618
Thymoquinone Inhibits Growth of Acute Myeloid Leukemia Cells Through Reversal SHP-1 and SOCS-3 Hypermethylation: In Vitro and In Silico Evaluation
I allow myself to make a series of suggestions separated into two categories.
Major corrections
- Is there evidence of any other molecules from natural products other than scutellarin that report effects similar to TQ? Lines 275-290 and 339-342.
- There is an error in the concentrations shown on lines 428 and 442.
Minor corrections
- Citation. Lines 20-21
- Correctly write IC50 subscript. Lines 92, 96, 102, 108, 119, 134, 413, and 441.
- Microsoft Excel does not calculate the IC50 it allows to represent and measure the SD. The IC50 determination comes from the assay. It would be much better to use statistical software such as Sigma-Plot or GraphPad Prism. Line 93. What type of treatment was used to calculate P? line 106.
- It is not justified to use Bisulfite-converted, change for bisulfite-converted. Lines 155 and 483
- The para position is italicized (descriptor) para. Line 234.
- Improve the resolution and orientation of the images in figures 8 and 9. The appreciation of the interactions is not the most adequate.
- The units of degrees Celsius are separated from the number, for example, 37 °C. Lines 390, 396, and 399.
- in table 5. Line 474.
- in table 6. Line 502.
- Check the correct writing of references 8, 21, 26, 31, 32, 35, and 61.

Author Response
Reviewer 2
Major corrections
Comment 1. Is there evidence of any other molecules from natural products other than scutellarin that report effects similar to TQ? Lines 275-290 and 339-342.
Authors Response:
Dear reviewer, evidence of the effect of curcumin, other molecule from natural products, on modulation of DNMT and JAK/ STAT pathway has added. Line 291-292 (reference 36). Line 343-345 (reference 46).
Comment 2. There is an error in the concentrations shown on lines 428 and 442
Authors Response:
Thank you for the comment and sorry for the typing error. The concentrations shown on lines 428 and 442 have rearranged correctly as shown in the manuscript track changes.
Minor corrections
Comment 3. Citation. Lines 20-21
Authors Response:
Thank you for the comment and please consider that lines; 20-21 are parts of the abstract, which should not be cited.
Comment 4. Correctly write IC50 subscript. Lines 92, 96, 102, 108, 119, 134, 413, and 441.
Authors Response:
Dear reviewer, your suggestion is highly appreciated. The subscript of “50” in IC50 has corrected in the whole manuscript.
Comment 5. Microsoft Excel does not calculate the IC50 it allows to represent and measure the SD. The IC50 determination comes from the assay. It would be much better to use statistical software such as Sigma-Plot or GraphPad Prism.
Authors Response
Thank you for the comment and please be informed that the IC50s were determined from the dose-cell viability % curve, which was blotted using Microsoft Excel.
Comment 6. It is not justified to use Bisulfite-converted, change for bisulfite-converted. Lines 155 and 483
Authors Response:
We highly appreciate your point of view and the comment has considered and “Bisulfite-converted” has been changed to “bisulfite-converted” as shown in the manuscript track changes.
Comment 7. The para position is italicized (descriptor) para. Line 234.
Authors Response:
Thank you for your comment. According to you suggestion we have italicized the “para” in line 239, as shown in the manuscript track changes. (Line No has changed because of editing in Figure 4)
Comment 8. Improve the resolution and orientation of the images in figures 8 and 9. The appreciation of the interactions is not the most adequate.
Authors Response:
Thank you for your suggestion, the resolution and orientation of the figures, 8 and 9 have improved.
Comment 9. The units of degrees Celsius are separated from the number, for example, 37 °C. Lines 390, 396, and 399.
Authors Response:
Thank you for your comment. The units of degrees Celsius in lines 402, 408, and 411 have corrected according to the comment. (Lines No have changed because of editing in Figure 4)
Comment 10. in table 5. Line 474.
Authors Response:
in Table 5. Line 487 has changed into table 5.
Comment 11. in table 6. Line 502.
Authors Response:
in Table 6. Line 515 has changed into table 6.
Comment 12. Check the correct writing of references 8, 21, 26, 31, 32, 35, and 61.
Authors Response:
Thank you very much and please be informed the writing of references 8, 21, 26, 31, 32, 35, and 61 has corrected. Reference 61 has changed to 63 because of adding two new references as suggested by reviewer 1.
